# The Performance Comparison of Socioeconomic and Behavioural Factors as Predictors of Higher Blood Lead Levels of 0–6-Year-Old Chinese Children between 2004 and 2014

**DOI:** 10.3390/children9060802

**Published:** 2022-05-30

**Authors:** Yixuan Xie, Yaohua Dai, Tao Li

**Affiliations:** 1Child Health Big Data Research Centre, Capital Institute of Pediatrics, Beijing 100020, China; xie1xuan@hotmail.com; 2Department of Integrated Early Childhood Development, Capital Institute of Pediatrics, Beijing 100020, China; cip_journal@163.com

**Keywords:** children, blood lead levels, China, predictors, lead exposure, sources

## Abstract

Childhood lead exposure is a commonly known risk factor affecting children’s health, and 10 governments have taken actions to reduce children’s lead exposure sources. Because lab testing for children’s blood lead levels (BLLs) was not popularized easily, socioeconomic and behavioural factors have been usually used as predictors of screening methods. Along with the overall decreasing trend of children’s BLLs, the lead-exposure-potential-predicting ability of such factors might be limited or changed over time. Our study aims to compare the predicting ability of multiple factors, including the living environment, economic disparity and personal behaviour differences between 2004 and 2014. With potential predicting factors identified, it could provide direction in identifying individual children facing high-risk lead exposure in the unit of clinics or communities of China. The study was first conducted in 12 cities in China in 2004 and then repeated in 2014 in the same 12 cities with the same method. In total, 27,972 children aged under 7 years were included in this study. With confounding factors adjusted, the child’s age, the family’s socioeconomic status and the child’s personal hygiene habit, especially biting toys, continued to be important predictors of higher blood lead levels among Chinese children. The sex of the child was no longer a predictor. Factors such as the father’s occupational contact with lead, residence near the main road and taking traditional Chinese medicine had the potential to be new predictors.

## 1. Introduction

Lead exposure impairs human’s health, especially children’s healthy growth and development [1,2,3]. Blood lead levels of children indicate when the children’s health would be harmed by the effects of lead in their inner system [4]. Blood lead levels of the Chinese children population have been decreasing due to the improvement of the environment and increasing social awareness on lead harm, but they are still much higher than those in developed countries [5,6]. In addition, no safe threshold for lead toxicity to human health has been clarified [3]. Blood lead laboratory testing is commonly used to determine blood lead levels [7]. However, the relatively limited capacity and the poor accessibility of laboratory testing make it difficult to meet the requirements of routine screening, especially in less developed countries, such as China. Therefore, an easier popularized and non-invasive pre-screening method is needed before sending the children into labs to test for accurate blood lead levels.

Demographic and socioeconomic factors, such as the sex of the child, parents’ education levels, the home construction time period, peeling paint or recent renovations and the child’s behavioural habits (e.g., pica, hand-mouth behaviour (often sucking fingers or chewing nails)), were usually used as predictors of elevated blood lead levels (BLLs) of children or to decide whether lab testing is necessary [5,8,9].

While the BLLs of children are decreasing around the world in the past few years, economic disparities are yet narrowing and the ability to predict lead exposure may be limited [10]. In China, the social environment has changed dramatically, yet the contribution of economic and behavioural factors to high lead exposure risk of children is not clear. In this paper, we examined the predicting ability of the living environment, economic disparity and personal behaviour in finding individuals with high-risk exposure in clinics or communities and assessed the variation of these factors’ predicting ability as they change with time.

## 2. Materials and Methods

### 2.1. Study Population

We conducted a cross-sectional survey in 12 cities in China (Xi’an, Guangzhou, Qingdao, Chengdu, Zhengzhou, Hohot, Hefei, Wuhan, Beijing, Changsha, Shijiazhuang and Yinchuan) from May to August 2004 and repeated the cross-sectional survey in the same cities again in 2014. Maternal and child hospitals/health centres in these cities were selected as study sites. After the recruiting started, all children aged less than 7 years and their caregivers were selected as participants when children went to the hospital or health care centre to receive vaccinations or physical examinations during this period (May to August) in each site. Children who were aged 7 years or older were excluded.

### 2.2. Questionnaire

After children’s caregivers provided written informed consent, trained doctors interviewed the caregivers face to face in their home language with a questionnaire. The questionnaire contained questions concerning demographic and socioeconomic information about the children, their parents and their home residence. The detailed information on the questionnaire is explained in our former paper [5]. This study was approved by the Capital Institute of Pediatrics Ethics Committee.

### 2.3. Blood Lead Concentration Laboratory Tests

A capillary blood sample (40 μL) was collected from each child participant by trained laboratory staff. After mixing with a diluent, the mixture was stored at 4 °C for laboratory analysis. All laboratories used by our study subscribed to the National System of External Assessment of the Quality of Results, which is conducted by the National Centre of Clinical Laboratories. A tungsten atomizer absorption spectrophotometer (Beijing Bohui Innovation Technology Co., Ltd., Beijing, China) was used in all the monitoring sites in these years. Detailed laboratory information is presented in another paper [5].

### 2.4. Statistical Analyses

Geometric means (GMs) and 95% confidence intervals of blood lead concentrations were used to report the average BLLs for both time points by selected factors. After the initial BLL data were transformed, the difference among the average BLLs was evaluated with the *t* test or ANOVA. Pearson’s chi-squared test was used for the proportion difference among levels of category factors. Least-squares means for factor combinations in a fitted linear model including sex, age, father’s education level and other related demographic and environmental factors were calculated. The *p* value was adjusted using the Bonferroni adjustment method to examine pairwise means comparison between every two levels of particular factors. Statistical analysis was performed using R statistical software, version 3.6.3 (R Foundation for Statistical Computing, Vienna, Austria).

## 3. Results

In total, 27,972 children aged under 7 years from two different time periods (2004 and 2014) were included in the analysis. The sex proportions of the two populations of children recruited from 2004 and 2014 were similar (*p* = 0.0799). In 2014, the number of children with fathers who were college graduates or beyond was 68.02%, higher than the 58.59% observed in 2004. Meanwhile, the number of children with fathers with an education level equal to or lower than high school graduation in 2014 was 31.98%, lower than the 41.41% observed in 2004. Therefore, in general, children’s fathers had higher education levels in 2014 than they did in 2004 (*p* < 0.0001); see Table 1. Children’s GM blood lead concentrations decreased from 46.76 μg/L (95% CI 46.21, 47.33) in 2004 to 33.37 μg/L (95% CI 33.05, 33.70) in 2014 (*p* < 0.0001). The percentage of children with blood lead concentrations equal to or higher than 50 μg/L in 2014 was 22.9%, less than half of that in 2004, which was 52.53% (*p* < 0.0001). The percentage of children with blood lead concentrations equal to or higher than 100 μg/L in 2014 was 1.14%, much less than that in 2004, which was 10.17% (*p* < 0.0001). The distribution of blood lead levels from these two time points is shown in Figure 1.

In 2004, female children’s adjusted GM of blood lead levels was 47.96 μg/L (95% CI 45.97, 50.03), while that of male children’s was 51.78 μg/L (95% CI 49.69, 53.97). In 2014, female children’s adjusted geometric mean of blood lead levels decreased to 42.71 μg/L, while that of male children’s decreased to 42.31 μg/L. Table 2 shows that blood lead concentration gaps induced by sex disappeared in 2014.

In 2004, older children (3–7 years old) had blood lead concentrations of 49.85–60.42 μg/L, higher than those of younger children aged under 3 years, which was 41.46–48.39 μg/L. In 2014, older children (3–7 years old) remained with higher blood lead concentrations (43.99–45.31 μg/L) than younger children (38.31–40.70 μg/L; *p* < 0.05), but the percentage change in 2014 was less than that in 2004. Participants whose fathers’ education level was at junior high school or lower had the highest GM blood lead concentrations at 50.89 μg/L in 2004 and 46.26 μg/L in 2014. In both 2004 and 2014, children’s blood lead concentrations increased as their father’s education level lowered.

Children’s GM blood lead concentrations increased if their family lived far from the main road to next to the street in both 2004 and 2014. In 2004, the adjusted GM value progressed gradually from 49.36 μg/L to 50.94 μg/L as the distance increased from residence to main road. After a decade, the adjusted GM value progressed from 39.30 μg/L to 43.12 μg/L as the distance increased from residence to main road. Although children’s BLLs decreased after 10 years, the change rate of the influence of distance from residence to main road on children’s blood lead concentrations increased significantly, from −0.97–3.20% to 7.10–13.69%. 

In 2004, peeled-off paint in the residence had little impact on children’s BLLs. Children’s GM blood lead concentration was 50.43 μg/L when there was no peeled-off paint from walls in their residence; when there was peeled-off paint from walls, their GM blood lead concentration was 49.24 μg/L, dropped by 2.36%. Children’s GM blood lead concentration was 49.59 μg/L when there was no peeled-off paint from furniture in their residence; when there was peeled-off paint from furniture, their GM blood lead concentration was 50.08 μg/L, increased by 0.99%. In 2014, children’s GM blood lead concentration was 42.09 μg/L when there was no peeled-off paint from walls in their residence; when there was peeled-off paint from walls, their GM blood lead concentration was 42.94 μg/L, with an increase of 2.02%. Children’s GM blood lead concentration was 41.72 μg/L when there was no peeled-off paint from furniture in their residence; when there was peeled-off paint from furniture, their GM blood lead concentration was 43.32 μg/L, increased by 3.84%.

Children’s GM blood lead concentrations increased as the frequency of having Chinese traditional medicine increased in both 2004 and 2014. In 2004, children’s adjusted GM BLLs increased from 49.01 μg/L to 50.92 μg/L as they ingested Chinese traditional medicine more frequently. After a decade, children’s adjusted GM BLLs increased from 39.92 μg/L to 46.63 μg/L as they ingested Chinese traditional medicine more frequently. Although children’s BLLs decreased after 10 years, the change rate of the influence of the frequency of taking Chinese traditional medicine on children’s blood lead concentrations increased notably, from 1.18–2.68% to 3.41–16.81%. 

In 2004, children’s GM blood lead concentration was 49.29 μg/L when their fathers’ occupation had no contact with lead, while it increased to 50.38 μg/L when their fathers’ occupation had contact with lead. In 2014, children’s GM blood lead concentration was 40.01 μg/L when their fathers’ occupation had no contact with lead, while it increased to 45.17 μg/L when their fathers’ occupation had contact with lead. The change rate of the influence of children’s fathers’ occupation having contact with lead on children’s blood lead concentrations increased from 2.21% to 12.90%. In contrast, the change rate of the influence of children’s mothers’ occupation having contact with lead on children’s blood lead concentrations did not increase much—from 2.05% to 3.40%. 

Whether children often sucked fingers or had pica did not affect children’s BLLs significantly in 2004, with a change rate of 0.62% and 1.66%, respectively, compared to children without these behaviours. However, the BLL change rate of having these two behaviours compared to not having these two behaviours increased to 4.45% and 4.89%, respectively, in 2014. The BLL change rate of children often biting toys compared to not biting toys was 7.81% in 2004, respectively. In addition, 10 years later, the BLL change rate of children often biting toys compared to not biting toys was 7.89%, indicating this behaviour still affected children’s BLLs. The details are shown in Table 2.

## 4. Discussion

Preceding petrol combustion and industrial emission lead deposited in surrounding soil, air and water [11]; residential products, such as peeling-off paint from furniture, walls and toys; and even some traditional Chinese medicine [12,13] are all potential sources of lead exposure for children. Testing children’s blood lead levels in laboratories is a common method to determine the lead concentration in children [7]. However, when the lab testing environment is not easily accessible, using predicting factors of potential lead sources might be a non-invasive and cost-effective way to narrow down the population of sensitive children who could be exposed to lead [14]. In this study, we aimed to determine and compare the performance of several socioeconomic and behavioural factors in predicting the population of children with higher lead exposure potential.

In urban areas of China, the father’s education level could represent the family’s socioeconomic status to a certain degree. Much of a family’s socioeconomic status information that usually is not easily obtained through questionnaires could be represented by the education level of family members. Blood lead concentrations were higher in children whose fathers had lower education levels, which showed that blood lead levels are easily affected by socioeconomic position. This situation was consistent with that in Korea and Europe [15,16]. In this study, as shown in Table 1, in general, children’s fathers had higher educational levels in 2014 than in 2004 (*p* < 0.0001). Although the father’s education level increased in general during this decade, it still influenced children’s BLLs. In Table 2, in both 2004 and 2014, children’s blood lead concentrations decreased as the father’s education level improved. Furthermore, although children’s BLLs decreased in general after a decade, the percentage change in children’s BLLs regarding their father’s education level increased from 2004 to 2014. Thus, the predicting ability of the father’s lower education level towards children’s higher BLLs not only did not disappear but also became more influential throughout the decade. With the father’s education level having more influence in determining children’s BLLs, it might indicate that the socioeconomic status of the family might become more important in predicting a high risk of lead exposure in Chinese children.

Previously, boys were more likely to have higher blood lead concentrations due to their negligence in keeping good personal hygiene habits compared to girls [6], which was consistent with data from the United States [17]. In this study, boys’ blood lead concentrations were 7.96% higher than those of girls in 2004, but then boys’ and girls’ blood lead concentrations became almost the same in 2014. Two reasons might explain this. First, the blood lead levels had been decreasing among the whole Chinese child population [18], which resulted in the difference in the value between boys’ and girls’ blood lead concentrations naturally becoming less. Second, in 2004, boys had relatively poor hygiene habits compared to girls and higher blood lead levels. After a decade, boys’ hygiene habits had improved overall, along with environmental development and better awareness throughout families in the society, and their blood lead levels became close to those of girls. Blood lead concentration gaps induced by sex disappeared in 2014. Thus, sex might not be used as a predictor of high blood lead levels anymore in the recent environment.

Older children were more likely to have higher blood lead concentrations than younger children [8,19]. Thus, it seemed that older age (≥3 years old) can be used as a predictor of high-risk lead exposure. This is because babies and toddlers had fewer self-activities, and they had little chance to be exposed to lead in their living environment. Meanwhile, frequent outdoor activities gave older children more opportunities to come in contact with lead contamination [6]. From 2004 to 2014, older children (3–7 years old) remained with higher blood lead concentrations than younger children (*p* < 0.05). The percentage of change in children’s BLLs increased as children grew older, from 15.36% to 45.73% in 2004. In addition, 10 years later, in 2014, although the percentage of change decreased to 4.88–18.27%, the change rate of children’s BLLs as they grew up was still non-negligible. Therefore, children’s age remained a steady predictor of children’s high-risk lead exposure potential.

The residential environment, either the outdoor environment or indoor conditions, contains a pool of lead. In 2004, as shown in Table 2, children’s blood lead concentrations did not change much as the children’s family residence was closer to the main road. This might be because children were exposed to various environmental lead sources, including lead in soil, and distance to the main road was not a predicting factor of potential lead exposure. Later, lead in soil was identified as a moderate contaminating metal in China [20]. Soil lead concentration was higher in main cities as compared to rural areas, which was associated with higher traffic density in large cities [21].

Soil containing lead might turn into dust when humidity reduces and then migrate into children’s residence with wind. There was a study that even when environmental lead sources (i.e., leaded petrol and household paint) reduced rapidly, high bioavailable lead concentrations in soil still existed along arterial roads and freeways [22]. In our study, in 2014, child’s blood lead concentrations would increase when the children’s family residence was closer to the main road. With other environmental lead sources treated effectively, lead in soil became a significant lead source for children. Therefore, after a decade, a shorter distance from residence to main road might have become a new predicting factor for children’s higher blood concentrations.

Paint is another important source of lead [23]. Globally, the production of lead-based paints is still widespread, and across Asia, Africa and Latin America, lead-based paints are still widely used in many developing countries [24]. There have been surveys indicating that paint containing a considerable amount of lead was being sold in China and a remarkable percentage of existing paint in kindergartens and primary schools had lead concentrations far higher than the national standard [25,26]. In this study, paint peeled off from walls or furniture in the residence was not significantly related with high GM blood lead concentrations of children in both 2004 and 2014, while in 2014, peeled-off paint in residences was related with higher unadjusted GM blood lead concentrations of children. However, after adjusted with the child’s sex, age, the father’s education levels and other relative factors, the increase rate of children’s GM blood lead concentrations when there was peeled-off paint in their residence was not statistically significant. Paint on toys was also considered in our study, and it will be discussed later. Here, according to the results, paint peeled off from walls or furniture in residences might not fit to be a predictor for high children’s blood lead concentrations.

Skin, hair, shoes and clothing of parents whose occupation had contact with lead could bring home lead dust [27]. As shown in Table 2, children whose father’s or mother’s occupation had contact with lead had higher GM blood lead concentrations than other children in both 2004 and 2014. After relative factors were adjusted, the increase in children’s GM blood lead concentrations due to their parents’ occupation having contact with lead compared to children with parents in occupations not having contact with lead became smaller. Notably, in 2004, the father’s and mother’s occupation having contact with lead had about the same influence on children’s BLLs, with a close increase in percentage. However, 10 years later, the father’s occupation had a greater impact on children’s BLLs than the mother’s occupation. The percentage of increase in children’s adjusted GM blood lead concentrations was 3.40% when their mother’s occupation had contact with lead, and it was 12.90% when the children’s father’s occupation had contact with lead. Therefore, whether the mother’s job had contact with lead was not a significant factor during the decade’s time of the survey, and it might be due to the mother’s better personal hygiene habits, such as changing clothes or taking a shower immediately after work. In addition, the actual numbers of mothers working in contact with lead was relatively lower than fathers. Only 5.48% of mothers’ occupations had contact with lead in 2004 and 3.63% in 2014, while 17.27% of fathers’ occupations had contact with lead in 2004, which dropped to 12.76% in 2014. From the data, we can see that as the overall trend of parents’ having jobs in contact with lead decreased over the decade, the percentage of mothers’ occupations in contact with lead was still a lot lower than that of the fathers’. The father’s occupation had become a potential predictor of high BLLs of children in 2014, partly because the fathers’ job choices were also linked to their education background. It was found in further analysis that 70% of fathers with an education background of college graduation or beyond had an occupation not in contact with lead, and only 28% of fathers with a college degree or beyond had an occupation in contact with lead in 2014. As analysed before, the father’s education level was significantly related with the child’s blood lead level during the decade. With a lower level of education, the father’s occupation choice was more likely to be with contact with lead. Therefore, comprehensively, if the fathers’ occupation had contact with lead, it could indicate high levels of blood lead in their children. It also indicated that promotion of healthier work environments and improvement of fathers’ health habits were still required.

Direct ingestion through the digestive tract is one of major pathways through which lead from the environment migrates into the children’s inner system. Children have a higher risk in this matter because of their more frequent hand-to-mouth behaviours and smaller body sizes compared to adults [28]. In this study, four questions were set to evaluate children’s personal hygiene habits: the frequency of the child sucking fingers, pica, biting stationery and biting toys. Sucking fingers is the most common hand-to-mouth behaviour among young children. The percentage of children often sucking fingers went down from 30.77% in 2004 to 27.65% in 2014. However, a statistically significant increase in the GM blood lead levels when children had the habit of often sucking fingers was only found among children in 2014, not in 2004. Although a 4.45% increase in BLLs did not necessarily mean that sucking fingers could be an influential predictor of lead ingestion, it still raised the alarm for parents that personal hand hygiene of children should be improved. The percentage of children often biting toys increased from 10.24% in 2004 to 16.98% in 2014. The increase in children’s GM blood lead concentrations when children had the habit of biting toys remained high during the decade—7.81% and 7.89% in 2004 and 2014, respectively. As a result, children’s toys remained a potential lead exposure source from 2004 to 2014, with multiple factors adjusted. As discussed above, lead-based paints are still widely sold in China, including paint on toys. A study in 2018 showed that a large number of painted toys sold in China had lead concentrations higher than Chinese standards [29]. The researchers purchased toys from one of the biggest online shopping platforms in China, and the result showed that approximately 12% of the toys contained paint, with total lead concentrations exceeding China’s regulatory standard for paints in toy manufacturing, and nearly 36% of the toys had lead concentrations exceeding the equivalent US regulatory standard and EU standard. The percentage of children having the habit of pica and biting stationery dropped significantly from 2004 to 2014, from 13.99% to 3.36% and 34.35% to 8.75%, respectively. The percentage change in adjusted GM blood lead concentrations when the child had these habits was not statistically significant. In general, from the results, it showed that parents had become more aware of the need to prevent children from developing habits such as biting stationery and pica that could expose children to lead ingestion. However, the percentage of children often biting toys did not change much from 2004 to 2014. This might be because parents were not aware that other than stationery, toys are a lead source as well, and they did not pay much attention to children’s behaviour such as biting toys. With toys containing lead-based paint widely sold in China, often biting toys continued to be a predictor of children with high BLLs, and it should be an important risk factor raising an alarm in next-step health education.

Altogether, the distance from the children’s family residence to the main road, the frequency of having Chinese traditional medicine, whether the father’s occupation has contact with lead and often sucking fingers were not significantly related with children’s higher GM blood lead concentrations in 2004, but they were significantly related with high BLLs in 2014. The older age of a child, the father’s education background and often biting toys remained related to higher GM blood lead concentrations of children throughout the decade from 2004 to 2014. The sex of the child was highly related to BLLs of children in 2004, but it was no longer related after 10 years’ time. Children with personal hygiene habits such as pica or biting stationery tended to have slightly higher blood lead concentrations, but these were not statistically significant after adjusting related factors.

Several limitations exist in this study. Firstly, a capillary sample rather than a venous blood sample was collected from each child enrolled to ensure a high participation rate in such a large sample. Secondly, limited environmental and socioeconomic factors were gathered; thus some valuable information might have been missed. Future studies can investigate children’s use of cosmetics, the household’s pipe water drinking condition, the frequency of traditional cookware usage, and leaded paint on the child’s bed. Finally, all the child participants were from cities and not towns or villages, which might have affected the accuracy of these potential predictors when it came to application in areas such as towns and villages.

## 5. Conclusions

The child’s age, the family’s socioeconomic status and the child’s personal hygiene habit, especially biting toys, continue to be predictors of high blood lead levels among Chinese children. The sex of the child is no longer a predictor. The father’s occupational contact with lead, residence near the main road and taking traditional Chinese medicine might be new potential predictors.

## Figures and Tables

**Figure 1 children-09-00802-f001:**
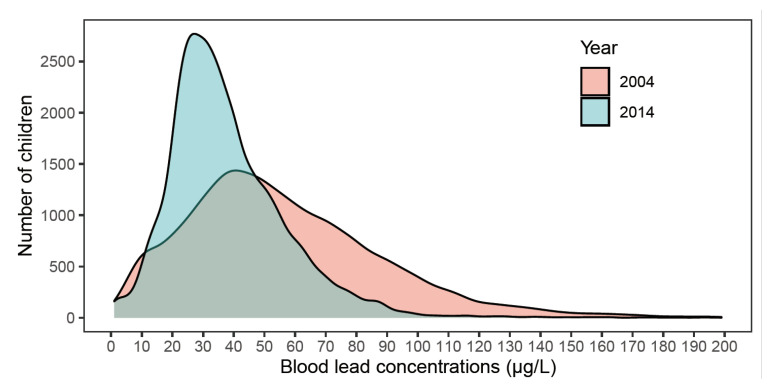
Distribution of blood lead concentrations of participants in 2004 and 2014.

**Table 1 children-09-00802-t001:** Demographics and blood lead levels of participants for two time periods.

	2004	2014	*p* Value ^#^
	N (%)	N (%)	
Total	13,852 (100)	14,120 (100)	
Sex			
Female	6173 (44.56)	6442 (45.62)	0.0799
Age			<0.0001
0–	1316 (9.50)	1366 (9.67)	
1–	1083 (7.82)	1397 (9.89)	
2–	1686 (12.17)	1363 (9.65)	
3–	3092 (22.32)	2679 (18.97)	
4–	2958 (21.35)	3181 (22.53)	
5–	2662 (19.22)	2723 (19.28)	
6–7	1055 (7.62)	1411 (9.99)	
Father’s education			<0.0001
College graduate or beyond	8026 (58.59)	9574 (68.02)	
High school graduate	4106 (29.98)	3327 (23.63)	
Junior high school or lower	1566 (11.43)	1175 (8.35)	
Blood lead concentration (GM (95% CI), μg/L)	46.76 (46.21, 47.33)	33.37 (33.05, 33.70)	<0.0001 *
Percentage ≥ 50 μg/L (%)	52.53	22.90	<0.0001
Percentage ≥ 100 μg/L (%)	10.17	1.14	<0.0001

^#^: Pearson’s chi-squared test. *: *t* test for log transformation of blood lead concentrations.

**Table 2 children-09-00802-t002:** Children’s blood lead levels (μg/L) by socioeconomic and behavioural factors for two time periods.

Factors	2004					2014				
		Unadjusted		Adjusted			Unadjusted		Adjusted	
	%	GM (95% CI)	Change, %	GM (95% CI)	Change, %	%	GM (95% CI)	Change, %	GM (95% CI)	Change, %
Sex										
Female	44.59	44.73 (43.93, 45.54) ^a^	-	47.96 (45.97, 50.03) ^a^	-	45.63	33.54 (33.07, 34.02) ^a^	-^.^	42.71 (40.87, 44.64) ^a^	-
Male	55.41	48.45 (47.68, 49.23) ^b^	8.32	51.78 (49.69, 53.97) ^b^	7.96	54.37	33.23 (32.79, 33.67) ^a^	−0.92	42.31 (40.52, 44.19) ^a^	−0.94
Age										
0–	9.50	37.52 (36.00, 39.11) ^a^	-	41.46 (39.13, 43.92) ^a^		9.67	30.22 (29.10, 31.39) ^a^	-	38.31 (36.28, 40.46) ^a^	
1–	7.82	44.85 (43.19, 46.57) ^b^	19.54	47.83 (45.09, 50.72) ^b^	15.36	9.89	31.69 (30.60, 32.83) ^a^	4.86	40.18 (38.13, 42.35) ^a^	4.88
2–	12.17	46.07 (44.59, 47.59) ^c^	22.79	48.39 (45.96, 50.95) ^b^	16.71	9.65	31.81 (30.80, 32.86) ^a^	5.26	40.70 (38.62, 42.90) ^a^	6.24
3–	22.32	46.68 (45.51, 47.88) ^c^	24.41	49.85 (47.60, 52.21) ^b^	20.24	18.97	33.89 (33.20, 34.61) ^a^	12.14	43.99 (41.99, 46.07) ^b^	14.83
4–	21.35	46.75 (45.54, 48.00) ^c^	24.6	49.40 (47.20, 51.70) ^b^	19.15	22.53	34.40 (33.80, 35.01) ^b^	13.83	45.15 (43.12, 47.28) ^b^	17.85
5–	19.22	49.73 (48.35, 51.15) ^d^	32.54	53.46 (51.03, 56.01) ^c^	28.94	19.28	34.00 (33.27, 34.74) ^b^	12.51	44.52 (42.50, 46.64) ^b^	16.21
6–7	7.62	56.65 (54.59, 58.79) ^e^	50.99	60.42 (56.98, 64.06) ^d^	45.73	9.99	35.35 (34.28, 36.46) ^c^	16.98	45.31 (43.11, 47.62) ^b^	18.27
Father’s education										
College graduate or beyond	58.59	45.32 (44.62, 46.04) ^a^	-	48.09 (46.07, 50.20) ^a^		68.02	32.24 (31.87, 32.62) ^a^	-	40.04 (38.32, 41.84) ^a^	
High school graduate	29.98	48.54 (47.47, 49.64) ^b^	7.11	50.57 (48.43, 52.80) ^b^	5.16	23.64	34.70 (34.03, 35.38) ^b^	7.63	41.49 (39.67, 43.40) ^b^	3.62
Junior high school or lower	11.43	49.36 (47.69, 51.10) ^b^	8.91	50.89 (48.36, 53.55) ^b^	5.82	8.35	39.38 (37.98, 40.85) ^c^	22.15	46.26 (43.93, 48.71) ^c^	15.53
Distance from residence to main road										
Far from the main road	35.89	46.24 (45.31, 47.19) ^a^	-	49.36 (47.24, 51.58) ^a^		38.78	31.40 (30.86, 31.94) ^a^	-	39.30 (37.56, 41.13) ^a^	
Separated by 2 buildings	17.84	45.75 (44.48, 47.05) ^a^	−1.06	49.41 (47.09, 51.85) ^a^	0.1	15.63	33.66 (32.91, 34.42) ^b^	7.2	42.09 (40.10, 44.17) ^b^	7.1
Separated by 1 building	14.90	46.06 (44.57, 47.59) ^a^	−0.39	48.88 (46.52, 51.36) ^a^	−0.97	19.77	34.46 (33.80, 35.13) ^bc^	9.75	43.59 (41.60, 45.67) ^b^	10.92
Separated by roadside green belt	13.68	47.06 (45.55, 48.61) ^a^	1.77	50.60 (48.10, 53.23) ^a^	2.51	12.22	35.38 (34.52, 36.27) ^c^	12.68	44.68 (42.52, 46.95) ^c^	13.69
Facing the street	17.69	47.91 (46.57, 49.29) ^a^	3.61	50.94 (48.62, 53.38) ^a^	3.2	13.60	35.60 (34.66, 36.57) ^c^	13.38	43.12 (41.10, 45.24) ^b^	9.72
Peeling-off wall and paint in the room										
No	80.53	46.70 (46.08, 47.32) ^a^	-	50.43 (48.34, 52.60) ^a^		84.20	32.97 (32.62, 33.32) ^a^	-	42.09 (40.28, 43.99) ^a^	
Yes	19.47	46.68 (45.35, 48.04) ^a^	−0.04	49.24 (47.11, 51.48) ^a^	−2.36	15.80	35.61 (34.80, 36.44) ^b^	8.01	42.94 (41.02, 44.96) ^a^	2.02
Peeling-off furniture paint in the room										
No	92.40	46.45 (45.86, 47.05) ^a^	-	49.59 (47.71, 51.54) ^a^		89.99	33.13 (32.8, 33.47) ^a^	-	41.72 (40.01, 43.51) ^a^	
Yes	7.60	47.79 (45.95, 49.70) ^a^	2.88	50.08 (47.55, 52.73) ^a^	0.99	10.01	36.49 (35.2, 37.82) ^b^	10.14	43.32 (41.15, 45.61) ^a^	3.84
Having Chinese traditional medicine										
Less than 2 times per year	66.43	46.20 (45.51, 46.90) ^a^	-	49.01 (47.08, 51.03) ^a^		78.61	32.66 (32.30, 33.02) ^a^	-	39.92 (38.27, 41.64) ^a^	
1−2 times per month	25.88	47.62 (46.52, 48.75) ^a^	3.07	49.59 (47.49, 51.78) ^a^	1.18	15.97	34.94 (34.14, 35.77) ^b^	6.98	41.28 (39.41, 43.24) ^b^	3.41
More than once per week	7.70	48.55 (46.59, 50.58) ^a^	5.09	50.92 (48.09, 53.91) ^a^	2.68	5.41	39.61 (38.00, 41.29) ^c^	21.28	46.63 (44.07, 49.34) ^c^	16.81
Father’s occupation having contact with lead										
No	82.73	46.24 (45.64, 46.86) ^b^	-	49.29 (47.24, 51.42) ^a^		87.24	32.59 (32.25, 32.93) ^a^	-	40.01 (38.18, 41.93) ^a^	
Yes	17.27	48.94 (47.52, 50.40) ^a^	5.84	50.38 (48.17, 52.70) ^a^	2.21	12.76	39.17 (38.18, 40.20) ^b^	20.19	45.17 (43.19, 47.25) ^b^	12.9
Mother’s occupation having contact with lead										
No	94.52	46.57 (46,00, 47.15) ^a^	-	49.33 (47.66, 51.05) ^a^		96.37	33.15 (32.82, 33.48) ^a^	-	41.81 (40.27, 43.41) ^a^	
Yes	5.48	48.95 (46.37, 51.68) ^a^	5.11	50.34 (47.42, 53.45) ^a^	2.05	3.63	39.61 (37.92, 41.38) ^b^	19.49	43.23 (40.62, 46.02) ^a^	3.4
Whether children often suck fingers										
No	69.23	46.74 (46.07, 47.43) ^a^	-	49.68 (47.65, 51.79) ^a^		72.35	32.99 (32.63, 33.35) ^a^	-	41.60 (39.79, 43.50) ^a^	
Yes	30.77	46.53 (45.54, 47.55) ^a^	−0.45	49.99 (47.89, 52.18) ^a^	0.62	27.65	34.40 (33.71, 35.10) ^b^	4.27	43.45 (41.58, 45.40) ^b^	4.45
Pica										
No	86.01	46.21 (45.61, 46.81) ^b^	-	49.42 (47.50, 51.42) ^a^		96.64	33.25 (32.93, 33.58) ^a^	-	41.51 (40.05, 43.02) ^a^	
Yes	13.99	49.77 (48.23, 51.36) ^a^	7.7	50.24 (47.88, 52.72) ^a^	1.66	3.36	37.16 (35.34, 39.06) ^b^	11.76	43.54 (40.93, 46.32) ^a^	4.89
Whether children often bite stationery										
No	65.65	45.28 (44.62, 45.95) ^b^	-	49.38 (47.25, 51.60) ^a^		91.25	33.07 (32.74, 33.41) ^a^	-	42.00 (40.25, 43.82) ^a^	
Yes	34.35	49.64 (48.61, 50.69) ^a^	9.63	50.29 (48.28, 52.39) ^a^	1.84	8.75	36.77 (35.61, 37.97) ^b^	11.19	43.04 (40.94, 45.24) ^a^	2.48
Whether children often bite toys										
No	89.76	45.96 (45.38, 46.55) ^b^	-	47.99 (46.18, 49.88) ^a^		83.02	32.83 (32.48, 33.18) ^a^	-	40.93 (39.15, 42.79) ^a^	
Yes	10.24	53.45 (51.57, 55.40) ^a^	16.3	51.74 (49.12, 54.51) ^b^	7.81	16.98	36.18 (35.35, 37.03) ^b^	10.2	44.16 (42.20, 46.21) ^b^	7.89

GM: geometric mean; CI: confidence interval. ^a^, ^b^, ^c^ ,^d^ and ^e^: the values with the same letter are not significantly different (*p* < 0.05). The *t* test for log transformation of blood lead concentrations was used, and the *p* value was adjusted using the Bonferroni adjustment method.

## Data Availability

Not applicable.

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
