# Peer review of "The Performance Comparison of Socioeconomic and Behavioural Factors as Predictors of Higher Blood Lead Levels of 0–6-Year-Old Chinese Children between 2004 and 2014"

_children, 2022, doi:10.3390/children9060802_

Round 1

Reviewer 1 Report

Below are some recommended revisions for the paper:

Line 10---Abstract: Childhood lead exposure is a commonly known risk factor affecting children’s health, and 10 governments have taken actions to reduce children’s lead exposure sources.

Line 23—Factors such as father’s occupational contact with lead, residence near the main road, and taking traditional Chinese medicine had the potential to be new predictors.

Line 30--- Blood lead levels of children indicate when the children’s health would be harmed by the effects of lead in their inner system [2].

Line 71-The detailed information on the questionnaire was explained in our former paper [19].

Line 86---After the initial BLL data were transformed, the difference….

Line 91-- P value was adjusted using Bonferroni adjustment method to examine pairwise means comparison between every two levels of particular factors.

Line 99-100--- In 2014, the number of children with fathers who were college graduates or beyond was 68.02%, higher than the 58.59% observed in 2004.

Line 101-102--- Meanwhile, the number of children’ with fathers with education level equal to or lower than high school graduate in 2014 was 31.98%, lower than the 41.41% observed in 2004.

Line 102-103-- Therefore, in general, children’s fathers had higher education levels in 2014 than they did in 2004 (P <0.0001).

Line 115-- Table 2 shows that blood lead concentration gaps induced by sex disappeared in 2014.

Line 121---…… but the percentage change 121 in 2014 was less than that in 2004.

Lines 194-197---- Preceding petrol combustion and industrial emission lead deposited in surrounding soil, air and water [17], residential products like peelings of furniture and walls, toys and even some traditional Chinese medicine [23, 29], are all potential sources of lead exposure for children.

Line 282--- … Notably, in 2004, father’s and mother’s occupation with contact of lead had about the same influence on…

Line 320---… sucking fingers could be an influential predictor of lead ingestion, it still raised the alarm….

Line 335-336---- The percentage change on adjusted GM blood lead concentrations when the child had the habits was not statistically significant.

Line 337-338----…. it showed that parents had become more aware of the need to prevent children from…..….

Line 342-----… much attention to children’s behaviour such as biting toys.

Line 344-345---- ….high BLLs, and it should be an important risk factor raising alarms in next-step health  education.

Line 359-362---- Future studies can investigate the child’s uses of cosmetics, the household’s pipe water drinking condition, the frequency of traditional cookware usage, and leaded paint on the child’s bed.

Line 368-370----- Father’s occupational contact with lead, residence near the main road, and taking traditional Chinese medicine might be new potential predictors.

Author Response

Response to Reviewer 1 Comments:

Line 10---Abstract: Childhood lead exposure is a commonly known risk factor affecting children’s health, and 10 governments have taken actions to reduce children’s lead exposure sources.

Thank you for your comment! I have revised the content according to your advice.

Line 23—Factors such as father’s occupational contact with lead, residence near the main road, and taking traditional Chinese medicine had the potential to be new predictors.

Thank you for your comment! I have revised the content according to your advice.

Line 30--- Blood lead levels of children indicate when the children’s health would be harmed by the effects of lead in their inner system [2].

Thank you for your comment! I have revised the content according to your advice.

Line 71-The detailed information on the questionnaire was explained in our former paper [19].

Thank you for your comment! I have revised the content according to your advice.

Line 86---After the initial BLL data were transformed, the difference….

Thank you for your comment! I have revised the content according to your advice.

Line 91-- P value was adjusted using Bonferroni adjustment method to examine pairwise means comparison between every two levels of particular factors.

Thank you for your comment! I have revised the content according to your advice.

Line 99-100--- In 2014, the number of children with fathers who were college graduates or beyond was 68.02%, higher than the 58.59% observed in 2004.

Thank you for your comment! I have revised the content according to your advice.

Line 101-102--- Meanwhile, the number of children’ with fathers with education level equal to or lower than high school graduate in 2014 was 31.98%, lower than the 41.41% observed in 2004.

Thank you for your comment! I have revised the content according to your advice.

Line 102-103-- Therefore, in general, children’s fathers had higher education levels in 2014 than they did in 2004 (P <0.0001).

Thank you for your comment! I have revised the content according to your advice.

Line 115-- Table 2 shows that blood lead concentration gaps induced by sex disappeared in 2014.

Thank you for your comment! I have revised the content according to your advice.

Line 121---…… but the percentage change 121 in 2014 was less than that in 2004.

Thank you for the advice. I put it as "but the percentage change in 2014 was less than that in 2004."

Lines 194-197---- Preceding petrol combustion and industrial emission lead deposited in surrounding soil, air and water [17], residential products like peelings of furniture and walls, toys and even some traditional Chinese medicine [23, 29], are all potential sources of lead exposure for children.

Thank you for your comment! I have revised the content according to your advice.

Line 282--- … Notably, in 2004, father’s and mother’s occupation with contact of lead had about the same influence on…

Thank you for your comment! I have revised the content according to your advice.

Line 320---… sucking fingers could be an influential predictor of lead ingestion, it still raised the alarm….

Thank you for your comment! I have revised the content according to your advice.

Line 335-336---- The percentage change on adjusted GM blood lead concentrations when the child had the habits was not statistically significant.

Thank you for your comment! I have revised the content according to your advice.

Line 337-338----…. it showed that parents had become more aware of the need to prevent children from…..….

Thank you for your comment! I have revised the content according to your advice.

Line 342-----… much attention to children’s behaviour such as biting toys.

Thank you for your comment! I have revised the content according to your advice.

Line 344-345---- ….high BLLs, and it should be an important risk factor raising alarms in next-step health  education.

Thank you for your comment! I have revised the content according to your advice.

Line 359-362---- Future studies can investigate the child’s uses of cosmetics, the household’s pipe water drinking condition, the frequency of traditional cookware usage, and leaded paint on the child’s bed.

Thank you for your comment! I have revised the content according to your advice.

Line 368-370----- Father’s occupational contact with lead, residence near the main road, and taking traditional Chinese medicine might be new potential predictors.

Thank you for your comment! I have revised the content according to your advice.

Reviewer 2 Report

    In this study correlations are made between factors concerning the living environment, economic disparity and personal behaviour, onee one hand and blood lead levels in children, in order to assess the predicting value of these factors in identifying individuals of high-risk exposure.

     The authors mention some limitations of the study. The conclusions are partially supported by results.

Minor issues:

- lines 48-49: I think that "their ability" instead of "...its ability..." would be correct;

- lines 194-197: please rephrase, because in its current form, the phrase is not very clear, in regard to its meaning;

- line 206: I think that "... usually is not...." instead of "...usually not..." would be correct;

- line 252: "as compared" instead of "comparing";

- line 337: "had become" instead of "had becoming";

Observations:

- a figure with the MDPI was probably misplaced in the manuscript above the title of Figure 1;

- In the manuscript it is mentioned that "The following supporting information can be downloaded at:  www.mdpi.com/xxx/s1, Figure S1: title; Table S1: title; Video S1: title.", but these are not available for download in the platform.

Author Response

Response to Reviewer 1 Comments:

Minor issues:

- lines 48-49: I think that "their ability" instead of "...its ability..." would be correct;

Thank you for your comment! I have revised the content according to your advice.

- lines 194-197: please rephrase, because in its current form, the phrase is not very clear, in regard to its meaning;

Thank you for your comment! I have revised the content according to your advice.

- line 206: I think that "... usually is not...." instead of "...usually not..." would be correct;

Thank you for your comment! I have revised the content according to your advice.

- line 252: "as compared" instead of "comparing";

Thank you for your comment! I have revised the content according to your advice.

- line 337: "had become" instead of "had becoming";

Thank you for your comment! I have revised the content according to your advice.

Observations:

- a figure with the MDPI was probably misplaced in the manuscript above the title of Figure 1;

This figure with the "MDPI" was pre-placed in the template. I'll consult with the editor where to place it properly.

- In the manuscript it is mentioned that "The following supporting information can be downloaded at:  www.mdpi.com/xxx/s1, Figure S1: title; Table S1: title; Video S1: title.", but these are not available for download in the platform.

These line were pre-placed in the template. I'm not sure if I can directly delete them. I'll consult with the editor for advice. Thank you again for your suggestions!